# Caregiving Role and Psychosocial and Individual Factors: A Systematic Review

**DOI:** 10.3390/healthcare9121690

**Published:** 2021-12-07

**Authors:** Elena Cejalvo, Manuel Martí-Vilar, César Merino-Soto, Marivel Teresa Aguirre-Morales

**Affiliations:** 1Department of Basic Psychology, Faculty of Psychology and Speech Therapy, Universitat de València, 46010 Valencia, Spain; elenacejalvoh4@outlook.es; 2Psychology Research Institute, Universidad de San Martín de Porres, Lima 34, Peru; 3Department of Psychology, Faculty of Psychology, Universidad Nacional Federico Villarreal, Lima 15088, Peru; maguirre@unfv.edu.pe

**Keywords:** psychological characteristics, physical disability, caregivers, systematic review

## Abstract

Taking care of a person with a physical disability can become a challenge for caregivers as they must combine the task of caring with their personal and daily needs. The aim of this study was to assess the impact that taking care of a person who needs support has on caregivers and to analyze certain characteristics they present, such as self-esteem and resilience. To that end, a bibliographic review was carried out from 1985, when the first article of taking care of a person who needs support was published, to 2020 (inclusive), in the databases of Web of Science (WoS), Scopus, Pubmed, Eric, Psycinfo, and Embase. The search yielded a total of (*n* = 37) articles subject to review, following the guidelines established in the PRISMA declaration. The results show that caregiving was highly overburdening and negatively affected the physical condition and the psychological and mental states of caregivers. In addition, certain psychological characteristics present in caregivers such as having high self-esteem and being resilient were found to act as protective factors against the caregiving burden.

## 1. Introduction

### 1.1. Historical Background on Care and Disability

Disability as a term was initially approached from three different perspectives: the dispensation model, the rehabilitative medical model, and the social model. The first model considered that disability had its origin in religious causes. Since disability was understood as a punishment or a divine curse, people with disabilities were socially excluded [1]. The medical model considered disability as a problem caused by a health condition that prevented the individual from coping with the demands of life in society. Due to that, the person needed to be treated in order to be able to cope with these demands [2]. Finally, the social model considered that the causes of disability were neither religious nor scientific in nature, but were largely social in nature, assuming people with disabilities should, and were able to, contribute and participate in society to the same extent as other people [3].

The inconsistency between the three models led to a new conceptualization of disability from a biopsychosocial perspective. In consequence, the World Health Organization [4] defines disability as deficiencies in human functioning that lead to limitations in activities and restrictions on participation. In this case, deficiencies refer to problems that harm certain body structures. The limitations are the problems that arise in the performance of a given task. Finally, the restrictions are the difficulties in taking part in a range of life situations. Disability is a complex problem which must be understood as an interaction between the characteristics of the human organism and the society in which it lives. Physical deficiencies, that is, those related to the body, limbs, and organs in general, are one of the disabilities with strong effects on human functioning.

Care, on the other hand, is a term associated with early civilizations, such as those of Sumer or ancient Egypt, where a family member was dedicated to the protection and maintenance of their family. This role was almost entirely assumed by women [5]. However, it was not until the late 1970s that sociology began to be concerned about the importance of taking care of people in a vulnerable situation, which implies giving greater visibility and importance to caregivers of this group, even though the role of caregiver had always existed [5]. However, caregivers, especially those providing informal care, are still coping with situations ignored by both the current social security system and society. The reason for this lies in the lack of an effective service for the strengthening of their skills and the recognition of their work.

The reality that encompasses the care of a dependent person and their respective caregivers is an issue that has been insufficiently investigated by the social sciences, though it is indisputably important today, reflecting demographic changes and transitions in markets and family models [2]. This suggests that caregivers as a group, who occupy a highly vulnerable place in our society, should be further investigated.

### 1.2. The Impact of Taking Care of a Person with a Physical Disability

Taking care of a person who is suffering from a physical disability can cause stress in caregivers, which may be associated with the interaction between the experience of burden and the perception of vulnerability to this burden [6]. There is a wide range of information available on what determines the burden on informal caregivers [7], which indicates a range of identifiable stressors that include: care permanence, dependency (physical or mental), the type and severity of the disease, the variety of roles, cognitive demands, and behavioral difficulties that occur between caregivers and the person they are taking care of Sandín (2003) stated that individual differences play a significant role in the stress process, which means that each person faces these demanding situations in a different way depending on the skills and the resources they have [8].

These stressors are not the only ones theoretically identified; however, their effects are related to a welfare reduction in the caregiver [9] and an increase in morbidity and mortality in stress operationalizations. 

Stress reactions associated with caregivers’ experience have been linked to factors such as the time spent in caring, the reduction in the caregivers’ social circle, the patient’s deterioration, the dependency on daily life activities, the recurrence of the disease, and the problematic behaviors that the person being cared for may present [10]. It is not difficult to recognize that caregivers as a population are at risk of developing a loss in their psychological well-being and an increase in stress-related illnesses.

As shown in a systematic analysis of 211 articles [11], neuropsychological assessments found that informal caregivers show symptoms of generalized cognitive impairment including decrease in memory, low selective attention and inhibition capacity, and deficiencies in the processing of verbal, visual, and digital information.

Another of the main consequences of care is the burden that caregivers experience [12]. The concept of burden dates back to the 1960s, when Grad and Sainsbury (1963) carried out a study to find out what effects caring for people with some type of psychiatric illness has on families [13]. This concept has been further researched and developed over the years. Caregiver burden is now defined as the caregiver’s subjective experience of care demands. Burden is a multidimensional concept that includes both objective and subjective characteristics. Objective burden refers to the changes the caregiver must make in various life domains, and subjective burden refers to the emotional reactions to the demands of caregiving [14].

The figure of the caregiver is crucial in the process of care, as the role is responsible for safeguarding and enhancing the self-esteem of the vulnerable group [15]. This role may be performed by professional caregivers or informal caregivers. The latter provide care to a person without any compensation or specific training [16] which can increase the experience of burden and vulnerability. The effect that training and skill acquisition have in caregiving is that of providing an individual caregiver with resources that can improve their capacity to cope with stress, in addition to improving the relationship between the caregiver and the person cared for, and the quality of life of both [15]. In the absence of training to improve caregiver skills and coping strategies, an informal caregiver is persistently vulnerable.

#### 1.2.1. Gender Differentiation

Apparently, another moderator that conditions the caregiver’s response to stress seems to be the gender of the caregiver. There is evidence suggesting that women tend to be more emotionally rooted in the role of caregiver, and that men tend to delegate more tasks, preserving their own well-being [17]. This gender differentiation has been maintained throughout history and, despite all the progress made in the area of gender equality, there are not many prospects for change in the short term, since the idea that women should be responsible for caregiving remains deeply entrenched in our society. Women tend to report a greater perceived burden associated with care, as well as more depressive symptoms, lower well-being, and worse physical health [18]. Among the potential stress experience moderators related to the care of older adults with disabilities, the sex of the caregiver and the informality of their role can be found.

#### 1.2.2. Protective Factors and Psychological Characteristics Present in Caregivers

For the caregiver and the person receiving care, there are also several protective factors that can mitigate the negative effects associated with care and provide better physical and mental health, as well as greater acceptance of disability [19]. Some of these protective factors are reciprocity within relationships, and a large social support network [20]. Social support becomes a necessity for caregivers because it is a resource that can provide information, training, and support, which can reduce burden and improve the process of acquiring caring skills [21]. In addition, there are certain psychological characteristics present in caregivers such as having high self-esteem or being resilient that can provide a high level of protection against their role assumption [22]. Resilience includes processes by which caregivers can continue their activity despite the occurrence of traumatic situations. Resilience depends as much on external factors as it does on internal factors, which interact with each other to enable development, despite the demands that this situation entails. There are certain factors that improve resilience such as social support, the coping styles used, or the caregiver assessment of stressors [23].

These protective factors are understood as influences that can modify or improve the caregiver’s response to a threatening situation that might, otherwise, create a non-adaptive outcome, and that can promote, therefore, well-being and satisfaction with the work that the caregiver performs.

### 1.3. The Current Study

An important implication, according to the literature reviewed in this study, is that caregivers of people with physical disabilities are exposed to recognizable challenges, such as coping with stress, increased costs, reduced employment opportunities, and reduced leisure time [24]. However, the accumulated research on the relationship between the caregiver and the mental health effects of care does not seem to have been synthesized.

Research synthesis is an approach that facilitates understanding the impact of care on caregivers’ lives, as well as the coping strategies and psychosocial factors involved. Synthesis research is characterized by a clear and orderly assessment of the literature through an explicit and concise research question and an appropriate evaluation of the evidence.

The aim of this study was, through a systematic review, to (a) determine the impact that the performance of the caregiving role generates on their psychological and physical health (impact here refers to the cognitive, emotional, attitudinal, and physical consequences linked to the exposure to the demands involved in the caregiver role), (b) identify psychological characteristics in caregivers (i.e., self-esteem and resilience), and (c) identify possible protective factors within the relationship between the caregiver and the person receiving care that mitigate the negative effects associated with care. The importance of this review also lies in the fact that the figure of the caregiver has been highly invisible for decades, even though they constitute the first resource of care for dependent persons and are responsible for safeguarding their lives. Therefore, it is essential to know the reality that exists around them, in order to offer resources that enhance their well-being and quality of life. 

## 2. Materials and Methods

This work is a systematic review of the scientific literature focused on the understanding of three aspects: (1) the impact that the performance of the role of caregiver generates on caregivers, (2) an analysis of certain characteristics of caregivers such as self-esteem and resilience, and (3) the identification of protective factors in the caregiver–person receiving care relationship, which can mitigate the negative effects associated with care. The PRISMA 2020 declaration guidelines [25] for systematic reviews were followed in order to achieve a correct formulation in this study.

### 2.1. Search

The current bibliographic review was conducted between January and June 2021 and was carried out in various different social and cultural contexts, registering studies from all countries, in order to enhance the generalization of the results. 

The procedure followed three steps. A first search iteration was carried out to obtain an overview of the topic; secondly, the application of the inclusion and exclusion criteria was carried out to identify and centralize the topic of work; and, lastly, a final search iteration was carried out manually to include those articles that had not been found after the first search iteration.

In the search sensitivity analysis approach, combinations of the terms psychological characteristics * AND caregivers * AND disability physical * were used to cover as many related works as possible. The databases in which the searches were conducted were Web of Science (WoS), Scopus, Pubmed, Eric, Embase, and PsycInfo. These databases were chosen because: (a) they allow obtaining an adequate and efficient collection of data, (b) they are the ones that show the vast majority of results, references, and access options in the field of research, (c) they are multidisciplinary databases, and (d) their content is examined by expert and qualified staff.

The articles included were those that (a) were published since 1985, the year in which the first article about caregivers of people in a dependent situation was published, until 2020 (both included), in order to cover as many articles as possible on the topic; (b) those that were published in peer-reviewed scientific journals; (c) regarding the time sequence, cross-sectional and longitudinal studies; (d) those that used quantitative, qualitative, and mixed approaches; (e) those containing results that could be quantified and generalizable; (f) those published in either English or Spanish; and (g) those with no limitation regarding the age of both people physically disabled and their respective caregivers (i.e., young people, adults, and the elderly). Articles published in other languages and/or typologies and which did not meet the requirements previously indicated were excluded from this review.

### 2.2. Eligibility

Prior to reading the abstracts and selecting the articles, the inclusion and exclusion criteria were defined. Articles that (a) examine the variables that are present in caregivers and have been related to the care of dependent people who suffer from some type of physical disability or some disease that affects their physical condition, (b) treat both informal and formal caregivers, (c) cover any age and sex range of both caregivers and disabled people, (d) have been published in a peer-reviewed scientific journal, (e) are original research, (f) have been published in both English and Spanish, (g) have open access, (h) use a longitudinal or cross-sectional approach (taking into account the time sequence) [26], (i) have a quantitative, qualitative, or mixed approach [27], (j) have been published between 1985 and 2020 (both included), and (k) follow any type of sampling (i.e., probabilistic or non-probabilistic) were included.

Regarding the exclusion criteria, (a) articles that were written in another language different from Spanish or English and other types of document types such as books or conference papers, (b) articles that did not talk about physical disability, (c) articles about caregivers without specifying the condition of the person receiving care, and (d) studies that had not been published and studies that had been published in other sources such as PhD theses and chapters of books were excluded.

### 2.3. Identification of Data Sources and Selection

Once the search was completed, a total of 383 articles were obtained: 97 articles were found in the WoS, 96 in Scopus, 91 in Pubmed, 5 in Eric, 59 in Embase, and 35 in PsycArticles. After the first search iteration was conducted, all the articles were downloaded and stored in the Mendeley Bibliography Manager and then analyzed in an Excel sheet. In the first instance, an initial screening was performed by reading the title and abstract of the articles previously indicated, in order to identify the relevance of the sample, the role of the caregiver, and the study of the impact and the factors associated with care. After that, duplicates were removed (*n* = 166), and the remaining 63 articles were read.

### 2.4. Data Extraction

After this first screening, a protocol was developed in a Microsoft Excel sheet to systematically extract the characteristics of the 63 studies that had been included in the review. For this purpose, the following characteristics were extracted: authors, year of publication, geographical location, type of sample, employment status, type of caregiver, type of physical disability, age of caregivers, gender of caregivers, sample size, instruments that were used to measure the variables, problems of bias of the selected sample, reliability of the instruments, main psychological characteristics present in caregivers, and other important study considerations and limitations.

### 2.5. Selection

Finally, an in-depth analysis of the full content of the 63 selected articles was carried out to assess their eligibility, and a total of 26 articles were removed because (a) they did not deal with physical disability (*n* = 13), (b) they did not address the psychological variables of caregivers (*n* = 4), or (c) the full article was not accessible (*n* = 9). The 37 articles that met the inclusion criteria described above were selected to carry out the systematic review. Figure 1 shows the screening process.

### 2.6. Methodological Limitations

However, the present study also has some methodological limitations:(a)Assessing the quality of the literature: The quality of each selected study was not assessed because the systematic review conducted was essentially descriptive and not evaluative. This was due to several reasons: First, it favored a large coverage of the studies retrieved and selected synthetic descriptions of the studies. Second, the selected studies led to the observation and highlighting of possible biases in them, such as the induction of the reliability and validity of the measures used, among other things noted above. These specific limitations would have been unnoticed if a filter was introduced on these aspects. This does not indicate the absolute absence of quality assessment of the studies because some assessment of this quality was partially guaranteed by the selection of the databases (e.g., WoS and Scopus), whose selective processes are high for choosing the journals receiving research articles.(b)The term psychological characteristics* suggests a wide range of psychological attributes linked to caregivers; however, there are many other attributes that could not be included in this review and, therefore, that are part of the methodological limitations of the current work. Despite this, it is important to note that other articles such as Berenguí et al. (2013) have also been published in high-impact journals and have used this combination of terms, considering the possibility that some psychological indicators were omitted [28].(c)Most of the studies analyzed do not report the sampling strategies used (i.e., probabilistic or non-probabilistic), the type of approach of the studies (i.e., quantitative, qualitative, or mixed), or the type of design they followed (i.e., cross-sectional or longitudinal).

## 3. Results

The synthesis of the results obtained in the different studies selected to carry out the review is shown in Table A1, which can be found in Appendix A. The table has been structured according to author and year, country, type of physical disability presented by the subject, type of sample, sample bias problems, instruments, type of caregiver, gender distribution, age of caregivers, employment status of caregivers, psychological characteristics of the caregiver, other important considerations of the caregiver, and limitations of the analyzed studies.

A summary of the articles selected from the systematic review is shown in Table 1.

### 3.1. Importance of the Physical, Social, and Cultural Context

This review includes studies that were carried out in different social and cultural contexts, in order to enhance the generalization of the results and to be able to observe the universality of the analyzed constructs. Constructs that influence care and its impact on caregivers such as stress, coping strategies, or cultural differences were studied. On this basis, a total of 33.3% of the studies were carried out in the USA, the country with the highest prevalence of studies related to caregivers of dependent people. On the other hand, the second country with the most studies related to this topic is Spain, with 11.11% of studies, followed by the United Kingdom (5.55%), Australia (5.55%), Switzerland (5.55%), the Netherlands (5.55%), and Saudi Arabia (5.55%). Finally, the countries where fewer studies have been conducted were Japan (2.77%), Canada (2.77%), Sweden (2.77%), India (2.77%), Israel (2.77%), Malaysia (2.77%), Turkey (2.77%), Norway (2.77%), and Italy (2.77%). Cultural processes often differ within the same ethnic or social group due to differences in age cohort, gender, class, religion, ethnicity, and even personality. In this way, different communication styles, decision-making preferences, or family roles can be appreciated among other factors. These factors produce variations which have been observed in the care of a dependent person and in their respective caregivers [29]. One of the studies [30] stated that Norway is considered one of the most gender-equal countries in the world. This differs greatly from the reality in other countries such as Israel, where all caregivers are women who are highly vulnerable due to the high burden associated with their job.

As it can be seen in the percentages of Table A1, most of the caregivers from the reviewed studies (94.60%) were informal caregivers. They have always been present in our society, although they are a highly invisible group. However, informal caregivers are currently the first assistance resource for taking care of people who need support [2].

Another factor taken into account in the studies was the age of the caregivers of a person with a physical disability. It was reported that most of them are adults (48.65%), and there is a lower prevalence of young (16.22%) and elderly (13.51%) caregivers. Age is a variable of important relevance and consideration. Different studies have shown that caregivers who are older (>65 years old) have poorer physical health and a greater burden associated with care [31,32]. Middle-aged women are the most frequent profile of informal caregivers within the family. In almost all age groups, there are many more female caregivers than male caregivers, and gender differences are particularly pronounced between the ages of 45 and 65. At these ages, there are up to six times more women than men caring for a dependent family member [33].

### 3.2. Methodological Aspects of the Analyzed Studies

We analyzed the time sequences of the studies. Both longitudinal and cross-sectional studies were taken into consideration in order to gather as much information as possible and to obtain differences in the results obtained by the two types of studies. A total of 56.76% of the studies were cross-sectional, where different variables were studied at a given time. As it is shown in the statistics, most of the studies included in this review follow this type of design. This type of design has many advantages, but it does not offer a cause–effect relationship, which may generate a certain bias in the information obtained. A total of 21.62% of the studies followed a longitudinal approach, which allowed data to be collected repeatedly over a certain period of time. This type of design helped to make observations and to detect any changes that occurred in the participants (i.e., caregivers or dependent people) of the studies.

We also analyzed sampling strategies. Most of the studies analyzed (59.46%) did not report sampling strategies. This is a methodological limitation since it does not allow us to know how the samples that are part of the studies and representative of the target population were obtained. Nevertheless, in 24.32% of the analyzed studies, non-probabilistic sampling was carried out, based on the subjective judgment of the experts. The types of non-probability sampling that were used in the studies are convenience sampling, quota sampling, and snowball sampling. On the other hand, 16.21% of the studies followed probability sampling using random methods. Regarding sample selection, some studies showed certain selection biases. The only bias reported was the loss of certain subjects during the study.

Finally, we analyzed the type of approach as another methodological aspect in the studies. However, most studies (64.86%) did not report them, which constitutes a methodological problem. Despite this, the approach that was carried out was quantitative in 13.51% of the studies, qualitative in 10.81%, and mixed in 10.81%.

### 3.3. Impact and Psychological Characteristics Present in Caregivers

Disability affects the process of personal development and is an experienced condition that produces changes at the personal, family, social, and cultural levels. Therefore, not only the person with disability is affected but also their environment, and within this, their caregivers, who develop their own perception of disability based on their own experience. As a consequence of the disability, the patient suffers a series of physical, cognitive, and emotional consequences that alter their functionality and autonomy and lead to a loss of independence that will directly influence their caregivers, affecting their physical, psychological, and mental states. However, there are also certain psychological characteristics present in caregivers (i.e., self-esteem and resilience) and a number of external factors that act as protective variables and mitigate the negative effects associated with care.

People who provide care to others are in a situation of need, as they require the necessary information and support during the process to achieve a greater quality of time, preserving their well-being and quality of life. It is therefore important to pay attention to caregivers’ physical and emotional health as this has the potential to influence the health, well-being, and successful rehabilitation of persons with disabilities [2,10].

The impact of taking care of a person with a physical disability on their respective caregivers is then presented, which translates into physical, psychological, emotional, social, and economic problems. Finally, the individual factors that are present in the caregivers and that favor this new care situation they have to face are presented.

It is important to note that the psychological consequences suffered by caregivers are interrelated, so the distinction into sub-themes has been made in order to present the information in a clearer and more concise way.

#### 3.3.1. Physical Problems Associated with Caregiving

The enormous investment of time and effort required to take care of a person in a situation of dependency produces a series of problems that affect the physical condition of their caregivers. This is especially true for people who suffer from some type of physical disability, who have often lost part of their mobility and need the help of another person to carry out their daily tasks. Two of the analyzed studies show that a large number of them had musculoskeletal discomfort, which causes problems associated with bone, joint, and muscle pain, as a result of the high demand required by these people. Activities such as bathing, personal hygiene, or changes in position generate physical problems for the family in charge, mainly in the musculoskeletal system, such as contractures, fractures, or back and spinal pain, which cause difficulties in well-being [34,35]. Other conditions associated with the physical burden were the fatigue and tiredness suffered by this group, due to the large amount of work involved in this role [36]. In addition, the presence of overload leads to variables that affect physical well-being because it requires effort and reduced sleep hours, which also lead to exhaustion, fatigue, and tiredness. Informal caregivers over the age of 65 are more vulnerable to injury. Additionally, only few of them receive training on how to perform these tasks efficiently and safely. Caregivers provide these services at home, non-medical settings that present their own challenges for care delivery, such as small and crowded spaces [36].

With regard to professional caregivers (e.g., nurses, rehabilitation staff), the transfers, liftings, and repositioning of patients are associated with musculoskeletal injuries. The exposure to these tasks is one explanation for the high rates of work-related musculoskeletal disorders occurring in healthcare workers [36].

Despite this, all the studies agree that the physical health of caregivers improved when the person they were taking care of had fewer symptoms associated with the disease. This fact reflects that the greater the demand required by the dependent person, the greater the physical burden on their support people [37,38].

#### 3.3.2. Psychological and Mental Problems Associated with Caring

##### Well-Being and Quality of Life

The psychological and mental states of caregivers are also affected by the patient demands and the loss of their autonomy. Three of the articles showed [36,39,40] that caregivers’ quality of life was reduced. Quality of life is a multidisciplinary concept, with great plasticity and subjective burden. For this reason, this construct is highly influenced by the caregiver’s evaluation of care, by their expectations, or by the degree of satisfaction in relation to care. Therefore, the singularities involved in caregiving, together with objective and subjective aspects, affect the quality of caregivers’ lives. 

In addition to quality of life, their psychological well-being was also attenuated. Psychological well-being is understood as the development of capabilities and personal growth, where caregivers show favorable indicators of functioning. Factors influencing the level of psychological well-being include the means employed to cope with the stress associated with the caregiving burden, i.e., the coping strategies used by caregivers and the caregiver’s subjective assessment of the resources available to them (e.g., perceived social support). Coping strategies are the cognitive and behavioral coping efforts developed to manage external and/or internal demands, which are considered to exceed the individual’s resources. These strategies are used to reduce the negative impact of stressors on psychological well-being. The use of some coping strategies facilitates adaptation and well-being. They also prevent illness and enable the informal caregiver to adapt to the deterioration of the person receiving care. Therefore, it has been shown that caregivers who do not use active coping strategies show poorer psychological well-being and poorer evaluation of the care they provide [9,30,41,42].

##### Emotional Stress and Psychological Distress

Despite the fact that emotional stress and caregiver overload appear in many articles as one and the same concept, we considered it appropriate to differentiate them in two sub-themes. Three articles [35,43,44] reported the emotional stress experienced by caregivers, in addition to the psychological distress they experience due to the provision of care [30,45,46]. The caregiver’s stress process is influenced by a number of interrelated elements, including primary stressors (related to the characteristics of the patient’s illness and the tasks performed by caregivers), and the caregiver’s subjective appraisal of these stressors, as well as their personal resources that may act as protective factors (e.g., social support and coping strategies). It is well established that taking care of a dependent person is a highly stressful experience. Caregivers’ stress is expressed in the form of family tension and pressure, which are experienced due to imbalances between carer responsibility and actual capacity. This generates a series of negative repercussions for the caregiver, mostly in those situations in which the caregiver perceives the situation as highly demanding, in which the care is provided in a continuous manner and the caregiver does not have the personal resources to cope with the caregiving situation adequately [47,48,49]. On the other hand, with regard to formal care, it is necessary to mention work-related stress, as numerous studies have shown its impact on psychological, physiological, organizational, and social variables that affect the individual, causing emotional and behavioral alterations. The impact of chronic stress on the physical and emotional health of formal carers has also been noted. The contact with their patients is often quite intense and sometimes not very rewarding as a great burden of responsibility is placed on the professional [47].

##### Anxiety and Depression

In addition to problems in well-being, quality of life, and stress, most caregivers tended to have a higher degree of anxiety [31,36,39] and greater depressive symptomatology compared to the rest of the population. Depression influences behavior, cognition, and emotions and significantly reduces the quality of attention. Depression is caused by negative self-evaluation, which makes it difficult to carry out daily routines as one feels fatigued and loses motivation.

Caregiver stress, social isolation, decreased leisure time, and lack of money are important indicators of depression and anxiety in caregivers [31,36,44,46]. In addition, caregivers of people with physical disabilities who provide 36 h of care or more per week have been found to be more likely than non-caregivers to experience depressive or anxious symptoms [42]. These conditions, significantly influenced by the absence of social support, put the psychological health of caregivers at risk [34,50,51]. Gender also plays an important role. As shown in two of the articles selected [10,31], women were more likely to suffer from depression compared to men. This may be due to the fact that women respond differently to all stages of the stress process than men, and they are more involved in and spend more time on caregiving tasks [42]. Finally, individual factors are also extremely important in the occurrence of mood-related problems, such as educational level, intercultural differences, or lack of personal resources of caregivers, which directly influence the occurrence of depressive and anxious symptoms.

##### Burden Associated with Caregiving

Zarit et al. (1980) defined the burden associated with caregiving as the degree of damage the caregiver suffers in terms of emotional and physical health, social life, and economic conditions as a result of taking care of family members [34]. The burden is made up of two components. On the one hand, there is subjective overload, understood as the psychological feeling associated with the fact of caring and, specifically, the emotional response to this experience. On the other hand, there is objective burden, which refers to the care tasks that the main caregiver must assume and the difficulties they have to face on a daily basis [34]. The burden associated with care was the most distressing factor for caregivers [39]. This burden was higher when disease symptoms were severe [9,34,36,43,52,53] and when reciprocity between patient and caregiver was low (i.e., when there was a potential give-and-take imbalance in the relationship) [54,55]. The burden was also related to the number of hours they provided care [39,56]. One aspect to be highlighted is that women are more involved in the care of dependent people than men [57]. As a result, women tended to have a higher burden of care and poorer mental and physical health [39,46,58,59]. Both being male and receiving external help in caregiving were positively related to caregiver satisfaction and lower caregiving burden [54].

#### 3.3.3. Other Problems Associated with Caregiving (Social, Family, and Financial)

The health and well-being of caregivers and the people they take care of were also affected by certain situations such as the existence of family conflicts, the presence of a hostile attitude, or the lack of warmth in the family nucleus [60,61]. Both hostile attitude and lack of warmth were measured through the Family Assessment Device (FAD) [62]. This tool examines the individual’s vision of their family relationships in terms of problem solving, communication, roles, affective response, affective involvement, behavioral control, and general functioning. As shown in two of the selected articles [45,63], when conflicts were solved through negative coping strategies, the nuclear family was affected. This was particularly the case when they were solved in an impulsive and careless way [64,65]. The way of coping with conflicts was also influenced by certain external factors such as tangible resources (education, money, standard of living, etc.), social support, or the existence of stressors (i.e., severity of illness). Taking care of these people also has a large impact on other areas of their caregivers’ lives. Eight of the studies [22,34,44,60,63,66,67,68] showed that coping with this situation implies that many caregivers have to leave paid jobs in order to devote themselves fully to care, thus facing significant financial problems, which directly affects their well-being. Many caregivers find it impossible to reconcile work and family due to the high demands placed on the person receiving care. This creates significant financial problems, as caring for a person with a disability involves many expenses and, in some cases, the financial support offered to families is scarce. As it is shown in one of the studies [44], when the financial barriers were fewer, caregivers had greater well-being.

#### 3.3.4. Individual and Protective Factors

Despite the great impact associated with care, there are also factors in caregivers that protect their health and well-being and promote the provision of care. As it has been shown in five of the articles that were selected for this review [11,52,54,60,69], the quality of life of caregivers was improved by three conditions: (a) the existence of a good relationship between the caregiver and the person receiving care, (b) having a broad support network [70,71] measured by the scale Perceived Social Support (MSPSS) [72], and (c) having positive personality traits or traits that are prone to adequate adaptation to their environment and effective coping responses, such as high self-esteem or resilience. While social isolation can be a stress factor, social support can act as a buffer against stress and, thereby, influence health and illness processes in different directions depending on its availability and appropriateness. On the other hand, self-esteem is one of the most important indicators in care, as having adequate self-esteem offers a willingness to cope with stress situations that come from their role as caregivers. Low self-esteem has been found to induce a cycle in which one is reluctant to ask for help, and to narrow social networks, which, in turn, exacerbate the burden of care. In contrast, caregivers with strong self-esteem experience less stress, as they are able to protect themselves from psychological challenges and to respond to situations in a more positive way. In addition to these factors, caregivers who resisted and recovered from threatening situations and high levels of stress (i.e., being resilient) experienced less anxiety, fewer health problems, and less negative affection. In addition, resilient caregivers have a sense of utility, of company, and of doing their duty and value the act of caring in a positive way. Being a resilient caregiver is an indicator of effectiveness in caregiving and is associated with positive coping. These are benefits that, in general, improve the caregiver’s quality of life [22]. Finally, those who felt free and secure in making decisions, and did not feel obligated, showed greater satisfaction with their lives and with providing care to the dependent person [70]. Despite the negative consequences outlined above, many caregivers think that providing care for a loved one has a positive impact on them and prefer to carry out this role themselves, rather than another professional.

Generally speaking, there are also positive aspects that caregivers often experience. In the case of informal caregivers, the time they spend with their relatives is greater, which makes them feel closer to this person. Secondly, caregivers feel useful as they are directly responsible for a person in a dependency situation, which increases their self-esteem, experiencing a possible personal growth. In addition, many caregivers report that through the act of caring, they have learned to be more understanding and patient, and to communicate greater affection to the dependent person.

## 4. Discussion

There has been little research conducted on the reality that refers to the act of taking care of a person with a physical disability. The figure of the caregiver has been unnoticed for many years, and it was not until the 1970s, with industrialization, that the needs and rights of this important group began to become visible in society [2].

With the presence of a person with a physical disability who needs to be taken care of, a new family situation occurs. It can generate a lot of changes that affect both the family structure and the condition of the people who take care of them, directly affecting their health to meet the burden and effort that the patient demands [73].

Most of the caregivers of these people are known as informal caregivers, who are typically part of the family itself and who have not received any previous training. However, there are also formal caregivers, who have received prior training and are paid for the work performed. The caregiver is a key figure in the care of a person in a vulnerable situation, as he or she is responsible for creating an individualized care plan based on the personal needs of the person receiving care.

Therefore, the aim of this review was to observe the impact that the performance of the caregiver’s role has on their psychological, physical, and social health, as well as the perception of their quality of life and the role of gender as variable modulators of these effects. The literature analyzed provides a framework to explain how variables impact health aspects of caregivers.

After the measurement of certain variables in caregivers, it was observed that the greatest affection suffered by them due to caregiving was the burden that this work produces, which affected their health and quality of life [39]. This burden was greater when the symptomatology of the disease was not favorable, when there was no reciprocity between the caregiver and the person receiving care, and when the number of hours spent taking care of these people was greater [9,34,36,43,52,53,54,55,56,66].

Studies that have measured caregiver burden, such as that of Blanco et al. (2019), show that observable differences in caregiver burden may also be due to certain cultural issues, which can influence how they perceive and respond to difficulties related to the care of a dependent person, although the tasks assumed by caregivers from different contexts are very similar. These cultural differences also have a great influence on the use of social support, reciprocity, support between the members of the family nucleus, or the way of expressing the emotions and feelings generated by both caregivers and the person receiving care [74].

Regarding the finding of Torgé (2014) on the effect of voluntary participation, it is highlighted that the perception of control of this new role of caregiving has a protective function against the perceived stress caused by it [70], because it can increase homeostasis between environmental demands and perceived well-being [75] and mitigate the negative consequences suffered by caregivers in developing their function [76]. Finally, it should be noted that perceived self-efficacy can also be a critical attribute for deciding on voluntary participation, as demonstrated in studies of different areas [77]. This also coincides with the study by Carpi et al. (2009), which also studied the differences between self-efficacy and the perception of control over the preventive behaviors of certain physical diseases. They concluded that self-efficacy significantly influences preventive behavior in a greater way than the perception of control [78,79].

The long-term care of a non-autonomous person also affects the well-being and quality of life of the caregivers, due to the high levels of responsibility and management burden during care, and because of the number of hours accumulated in daily activities [9,30,36,39,40,41,42]. This situation coincides with the finding of Flores et al. (2014), who stated that taking care of a dependent person is a stressful experience that can become chronic and have a series of negative effects for the caregiver, such as a reduction in their well-being. This is especially clear in cases where caregivers perceive the situation as highly demanding and the act of caregiving takes place on a continuous and prolonged basis [47].

Sometimes, the depressive or anxious symptoms presented by caregivers are added to these situations due to the continuation of providing care for these people, which affects their emotional health and is directly related to the expression of their emotions and their intrapersonal relationships [31,36,39,44,46]. In addition, it has also been observed that suffering from anxiety and depression causes, as a result, an increased risk of suffering from other problems that affect physical health such as fatigue, tiredness, or musculoskeletal discomfort [34,35,36,67].

Based on the above, as shown by Gálvez et al. (2003), there has been an increase in the prevalence of mood disorders among caregivers of dependent people in recent years, especially an increase in the frequency of anxiety and depression problems. It has been found that the higher the degree of dependency, the higher the prevalence of affective disorders among caregivers [80].

From everything previously discussed, it is suggested that caregivers are in a situation of constant exposure to stressors, whose influence on physical and psychological effects is well known. The impact of a disability is a distressing experience where preconceived schemes and emotional responses of the caregiver will also come into play. In addition, all of these conditions occur significantly more in caregivers than in the general population.

In addition to the physical and mental impact of caregiving, the studies analyzed also showed the gender differences that exist in caregivers, with women assuming the majority of care, taking into account not only the burden but also how responsibilities are divided, showing worse self-informed health compared to men and spending more time in care [10,31]. This often means that they have to reconcile work and family life or, in the worst case, abandon their professional life in order to devote themselves fully to the care of these people [22,34,44,60,63,66,67]. The results of the 7th Survey on Quality of Life at Work of 2006 carried out by the Spanish *Ministerio de Trabajo* showed that women were less satisfied than men regarding the level of balance achieved between family and work. Having to spend most of their time on the care of another person, where, in many cases, the work–family balance is not viable, means a considerable reduction in their well-being, due to all the problems that this entails [44].

Despite this, other protective variables that decrease the negative effects that the care of a dependent person has on the health of their caregivers were also analyzed. These protective variables are constituted by certain individual characteristics of caregivers and by external factors, which facilitate the provision of support to other people. Among the protective factors within the caregiver–person receiving care relationship, we can find: the existence of a good relationship between the two, a support network to be able to count on and receive help, or having good self-esteem and being resilient [11,52,54,60,69]. These factors mitigate the negative effects described above.

These psychological characteristics presented in caregivers play a crucial role in care, since, as shown by Useros et al. (2010), high self-esteem has been linked to the use of effective coping strategies, good acceptance of disability, and high satisfaction with care [81].

Other authors such as Carvalho et al. (2006) affirmed that resilience is an important protective factor because they found that being a resilient person is an indicator of effectiveness in the care provided. Resilient people showed less exhaustion and more positive coping strategies, which contributed to improving their quality of life [82].

On the other hand, regarding the methodological aspects of the selected studies, there is a non-trivial amount of lack of information in the research methodology used, which includes the type of sampling, the reliability of the scores, the detection of significance, the treatment of neglected or insufficiently strained responses, and the corroboration of the validity of the internal structure of the instruments used. Missing information is not a good reporting practice and can be pointed out as a problem that involves not only authors but peer review. This problem of disinformation by omission is defined by the so-called induction of reliability [83] and induction of the validity of measures [84,85,86], but can be generalized to any information relevant in the methodology used.

Finally, Useros et al. (2010) addressed the possible effect of social support as an important buffer to the stress effect on caregivers. In addition, the influence that social support had on the health of caregivers has been demonstrated, showing that caregivers who received more support had fewer depressive symptoms [81].

### 4.1. Practical Implication

Systematic reviews provide viable recommendations for further studies on caregivers living with persons with physical disabilities. An example of this would be the development of psychosocial interventions whose objectives are feasible to be implemented in families, interpersonal relationships, and society. These interventions could reduce the levels of stress, anxiety, and emotional distress, as well as improving intra-family and interpersonal relationships and the quality of life.

These analyses show key aspects to respond to new problems or to deepen aspects that are still poorly investigated. This procedure will also lead to an analysis from different angles that will, in turn, increase the understanding of the problems presented. Thanks to this, a new level of confidence can be reached according to authors, methodologists, and specialists.

### 4.2. Limitations and Future Research

The interpretation and implications of this study should be framed by its limitations. First, the size of the selected studies was generally small, and this introduced significant variations due to the size of the sampling error. Second, no other variables were measured that could be of interest for the results reported by the different studies analyzed: for example, empathy, since some studies considered empathy as a protective variable for the physical and mental health of caregivers [87]. Third, there were differences in the characteristics of caregivers of people with physical disabilities, which limits a safe generalization of the results obtained, such as the age of the participants, the type of caregiver, the gender distribution of the participants, or the nationality. Specifically, there was a large difference in the gender of the caregivers, with the majority of women assuming this role. Fourth, the caregiver role considers both formal and informal caregivers; however, the vast majority of the studies reviewed were oriented towards informal caregivers, meaning there may be restrictions on the generalization of the effects encountered towards formal caregivers. Fifth, this review was restricted to articles published in a peer-reviewed scientific journal, excluding studies that had not been published and studies that had been published in other sources such as PhD theses or book chapters. Sixth, as previously indicated, the term psychological characteristics* suggests a wide range of psychological attributes linked to caregivers; despite this, there are many other attributes that could not be included in this review, since only self-esteem and resilience appear as individual factors that act in a protective way in the care of these people. Seventh, the bibliographic search was carried out by a single researcher, meaning the estimation of inter-judge reliability could not be measured.

The current situation shows that the care of a dependent person produces a number of consequences in their support figures. Due to this, issues aimed at the development of the caregivers’ welfare, with emphasis on improving their quality of life, should be investigated in future research. With regard to psychosocial factors, it is of vital importance to delve into the protective factors present in caregivers of dependent people, since those shown in this study, such as having a broad support network, presenting positive personality traits such as good self-esteem, or having adequate coping strategies, have been shown to mitigate the negative effects associated with care. On the other hand, regarding individual factors, it would be of great interest for future lines of research to be more targeted at male caregivers, since, as this study shows, gender stereotypes that place women as the main figures in care are still present in our society. This would allow us to observe the effects that the care of a person with disabilities has on them, and to carry out a comparison with the effects that have been visualized throughout this study that mainly concern women. With regard to the health of caregivers, as has been observed throughout this work, both physical health and psychological health are strongly affected by the performance of this role. In the future, therefore, apart from promoting more effective and preventive programs to provide resources for caregivers, it would be essential to look further into the psychological effects this work has on them, in order to safeguard their health to the fullest extent and to provide them with sufficient resources and tools to deal with their role. It may also be considered to recognize the difficulties that would be associated with the caregiver in terms of lifestyle, family, society, education, and health. It is also necessary to know what model of disability is perceived by caregivers, who can be formal or informal caregivers, in order to understand how their mediating variables are activated as a process of prevention in their health, considering their cognitive patterns, attitudes, and behaviors in the daily assistance of dependent people.

## 5. Conclusions

Taking into account the results shown in this systematic review, taking care of a person with physical disabilities has a great impact on their caregivers, generating negative effects in their physical condition and in their mental and psychological states. It is important to note that the differences related to the act of care occurring nowadays place women in a situation of greater vulnerability, exposing them to worse consequences associated with care, such as greater burden or greater depressive symptoms. However, certain characteristics present in caregivers, such as maintaining good self-esteem or being resilient, or linked to other external factors, such as having a broad support network, have been found to mitigate the negative effects associated with care. Despite this, the reality of taking care of a person with a physical disability has been little investigated, and it is vitally important to continue deeply exploring this topic.

## Figures and Tables

**Figure 1 healthcare-09-01690-f001:**
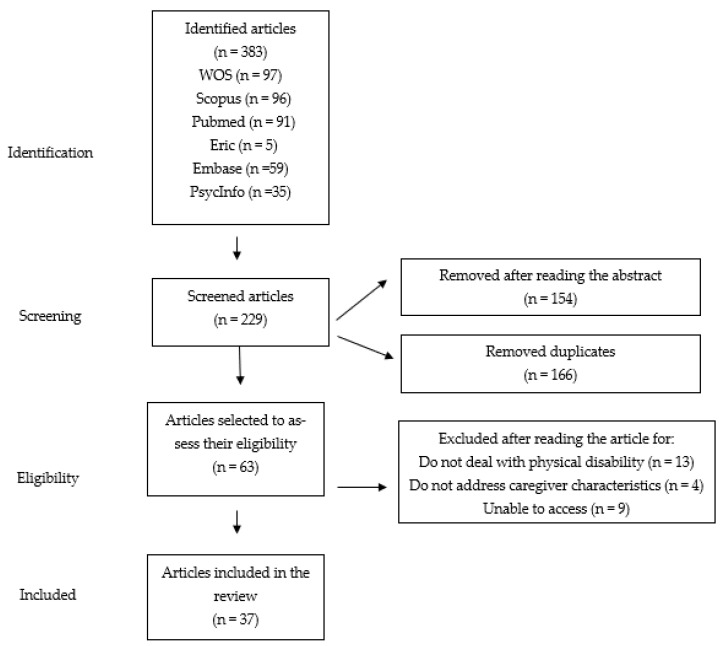
Flowchart according to PRISMA.

**Table 1 healthcare-09-01690-t001:** Summary of selected articles.

Main Characteristics	*n*	%	Studies
Type of caregivers			
Formal caregivers	2	5.40%	13, 32.
Informal caregivers	35	94.60%	1, 2, 3, 4, 5, 6, 7, 8, 9, 10, 11, 12, 14, 15, 16, 17, 18, 19, 20, 21, 22, 23, 24, 25, 26, 27, 28, 29, 30, 31, 33, 34, 35, 36, 37.
Sample type (people receiving care)			
Young people	6	16.22%	7, 12, 15, 20, 27, 30.
Adults	18	48.65%	1, 3, 5, 8, 11, 13, 16, 19, 21, 23, 24, 25, 26, 28, 31, 33, 35, 37.
Elderly	5	13.51%	6, 9, 14, 18, 32.
All ages	5	13.51%	10, 17, 22, 29, 36.
NR	3	8.11%	2, 4, 34.
Distribution by sex (caregivers)			
Men	1	2.70%	25
Women	5	13.51%	9, 17, 27, 30, 37.
Both	19	51.35%	1, 2, 3, 6, 8, 10, 13, 14, 15, 18, 19, 21, 22, 28, 29, 32, 34, 35, 36.
NR	12	32.43%	4, 5, 7, 11, 12, 16, 20, 23, 24, 26, 31, 33.
Type of design			
Cross-sectional	21	56.76%	5, 6, 11, 12, 13, 14, 15, 17, 18, 19, 20, 21, 22, 23, 25, 27, 29, 34, 35, 36.
Longitudinal	8	21.62%	3, 9, 10, 16, 24, 28, 30, 17.
NR	8	21.62%	1, 2, 4, 7, 26, 31, 32, 33
Sampling strategies			
Probabilistic	6	16.21%	6 (SS), 9 (SRS), 24 (SRS), 19 (SS), 20 (SS), 22 (SS)
Non-probabilistic	9	24.32%	1 (CS), 2 (QS), 3 (QS), 8 (QS), 11 (SBS), 12 (SBS), 13 (CS), 15 (QS), 16 (QS)
NR	22	59.46%	4, 5, 7, 10, 14, 17, 18, 21, 23, 25, 26, 27, 28, 29, 30, 31, 32, 33, 34, 35, 36, 37
Type of approach			
Quantitative	5	13.51%	3, 6, 21, 27, 29
Qualitative	4	10.81%	8, 11, 32, 35
Mixed	4	10.81%	1, 7, 16, 37
NR	24	64.86%	2, 4, 5, 9, 10, 12, 13, 14, 15, 17, 18, 19, 20, 22, 23, 24, 25, 26, 28, 30.31, 33, 34, 36

SS (stratified sampling); SRS (simple random sampling); CS (convenience sampling); QS (quota sampling); SBS (snowball sampling); NR (not reported).

## Data Availability

The analysis script is available on request from the authors.

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
