# Peer review of "Caregiving Role and Psychosocial and Individual Factors: A Systematic Review"

_healthcare, 2021, doi:10.3390/healthcare9121690_

Round 1

Reviewer 1 Report

The manuscript addresses a topic of interest to the international scientific community, such as the impact associated with caregiving and the variables present in caregivers of people with physical disabilities. The following recommendations are made below to improve the evaluated text:

TITLE: The title does not clearly reflect the content of the manuscript. It is recommended to specify that it is a Systematic Review. It is suggested as a title “Impact and variables associated with the role of a caregiver of a person with a physical disability: Systematic Review”.

ABSTRACT: It is recommended to explain the methodology and the main results more clearly.

INTRODUCTION: The study does not provide sufficient background. It is suggested to include previous studies on the subject based on biographical reviews, specifying their contributions and limitations for the present investigation.

METHODOLOGY: The materials and methods used are not adequately described. It is necessary to specify the analysis of the selected literature based on: location of the studies (importance of the physical, social and cultural context on the results), approximation of the studies (cross-sectional and longitudinal), experimental studies, quantitative approaches, qualitative approaches, problems of bias in the selection of the sample and the information of the studies, etc. Also, it is important to explain the methodological limitations. In addition, it is suggested to indicate the software used.

RESULTS: Results are not clearly presented, so this section should be revised. There is no adequate analysis of the results of the selected studies. It is recommended to analyze the data: location of the studies (importance of the physical, social and cultural context on the results), approximation of the studies (cross-sectional and longitudinal), experimental studies, quantitative approaches, qualitative approaches, bias problems in the selection of the sample and the information of the studies, etc.

DISCUSSION AND CONCLUSIONS: There is a limited theoretical discussion based on the results obtained in the study. It is recommended to discuss the results from the theory. Also, it is suggested to improve the conclusions of the study, as well as to propose possible lines of research.

Author Response

Respuesta a los comentarios del revisor 1

Se realizaron cambios importantes para mejorar sustancialmente el manuscrito, y agradecemos al revisor sus entusiastas observaciones. Creemos que estos cambios se aplicaron con la minuciosidad que esperaba el revisor, y nuevamente agradecemos al revisor por el trabajo de revisión.

Revisor 1: El manuscrito aborda un tema de interés para la comunidad científica internacional, como es el impacto asociado al cuidado y las variables presentes en los cuidadores de personas con discapacidad física. Las siguientes recomendaciones se hacen a continuación para mejorar el texto evaluado.

Autores: Nos gustaría agradecerle su trabajo y su interés en este estudio.

Point 1: The title does not clearly reflect the content of the manuscript. It is recommended to specify that it is a Systematic Review. It is suggested as a title “Impact and variables associated with the role of a caregiver of a person with a physical disability: Systematic Review”.

Response 1: Thank you for your appreciation of the title, we agreed to change it to make it clearer and to reflect the content of the manuscript.

Self-esteem, resilience, and impact on caregivers of a person with physical disability. Systematic Review.

Point 2: ABSTRACT: It is recommended to explain the methodology and the main results more clearly.

Response 2: Based on your feedback, your comments have been taken into account. The methodology used was described and the main results found in this study were included.

               “Caring for a person with a physical disability becomes a challenge for caregivers, as they have to combine the task of caring with their personal and daily needs. The aim of the study was to analyze the impact that caring for a dependent person has on caregivers and to analyze certain characteristics, such as self-esteem and resilience, that are present in the caregivers of these people. To this end, a bibliographic search was carried out from 1985, the year in which the first article on caring for a dependent person was published, until 2020 (both included), in the databases of Web of Science (WoS), Scopus, Pubmed, Eric, Psycinfo and Embase, yielding a total of (N=37) articles subject to review and following the guidelines established in the PRISMA 2020 declaration. The results showed that caregiving was highly overburdening and negatively affected the physical condition and the psychological and mental state of caregivers. In addition, certain psychological characteristics present in caregivers, such as having good self-esteem and being resilient, were found to act as protective factors against caregiving burden”.

Point 3: The study does not provide sufficient background. It is suggested to include previous studies on the subject based on biographical reviews, specifying their contributions and limitations for the present investigation.

Response 3: As for the suggestions in the introduction, new contributions of background studies based on bibliographical references were incorporated, we have considered the respective limitations at all times.

For example:

               1.1 Historical background on care and disability.

Disability is a term that was initially approached from three different conceptions: the model of dispensation, the rehabilitative medical model, and the social model. The first model considered that disability had its origin in religious causes, the exclusion of these people was the primary characteristic projected from this model, since disability was understood as a punishment or a divine curse [1]. For its part, the medical model considered disability as a problem caused by a health condition that prevented the individual from coping with the demands of life in society, so that the person had to be treated in order to be able to cope with these demands [2]. Finally, the social model considered that the causes of disability were neither religious nor scientific in nature, but were largely social in nature, assuming that these people should and were able to contribute and participate in society to the same extent as other people [3].

               1.2 The impact of caring for a person with a physical disability.

The concept of burden dates back to the 1960s, when Grad and Sainsbury (1963) carried out a study to find out what effects caring for people with some form of psychiatric illness had on families [13]. This concept has been further researched and nuanced over the years. Caregiver burden is now defined as the caregiver's subjective experience of care demands. Burden is a multidimensional concept that includes both objective and subjective characteristics. Objective burden refers to the changes the caregiver must make in various life domains and subjective burden refers to the emotional reactions to the demands of caregiving [14].

               1.2.2 Protective factors and psychological characteristics present in caregivers.

Social support becomes a necessity for caregivers and constitutes a resource that offers information, training and support to the caregiver, which decreases caregiver overload and enhances the process of acquiring caregiving skills [21]. In addition, there are certain psychological characteristics present in caregivers such as having high self-esteem or being resilient, which are highly protective against taking on the caregiving role [22]. Resilience comprises a process that allows caregivers to resume their development despite the occurrence of a given traumatic situation. Resilience depends on both external and internal factors, which interact with each other to provide continuity of development despite the circumstances of the situation. There are certain conditions that drive resilience, such as social support, coping styles used or the assessment of stressors [23].

Point 4: The materials and methods used are not adequately described. It is necessary to specify the analysis of the selected literature based on: location of the studies (importance of the physical, social and cultural context on the results), approximation of the studies (cross-sectional and longitudinal), experimental studies, quantitative approaches, qualitative approaches, problems of bias in the selection of the sample and the information of the studies, etc. Also, it is important to explain the methodological limitations. In addition, it is suggested to indicate the software used. 

Respuesta 4: Basado en su observación, se adaptó la descripción de materiales y métodos. En primer lugar, se localizaron los estudios, se indicó también el tipo de abordaje y las estrategias de muestreo (Cuadro 2). Finalmente, se indicaron las limitaciones metodológicas (2.6) y se indicó el software utilizado.

Punto 5: Los resultados no se presentan claramente, por lo que esta sección debe revisarse. No existe un análisis adecuado de los resultados de los estudios seleccionados. Se recomienda analizar los datos: ubicación de los estudios (importancia del contexto físico, social y cultural en los resultados), aproximación de los estudios (transversal y longitudinal), estudios experimentales, enfoques cuantitativos, enfoques cualitativos, problemas de sesgo en la selección de la muestra y la información de los estudios, etc.

Response 5: The results were expanded. We included the location of the studies, we also added the methodological aspects (the time sequence of the studies, the sampling strategies, and the type of approach). The results concerning the impact and psychological characteristics of caregivers were theoretically supported.

Point 6: There is a limited theoretical discussion based on the results obtained in the study. It is recommended to discuss the results from the theory. Also, it is suggested to improve the conclusions of the study, as well as to propose possible lines of research. 

Response 6: The discussion was supported by methodological aspects with respect to results found in other studies.

 For example:

“Esta situación coincide con el hallazgo de Flores et al. (2014), quienes afirman que cuidar a una persona dependiente es una experiencia estresante que puede volverse crónica y tener una serie de efectos negativos para el cuidador, como una reducción de su bienestar. Esto es especialmente evidente en los casos en que los cuidadores perciben la situación como muy exigente y la atención se realiza de forma continua y prolongada [47] ”.

También hicimos cambios a las conclusiones y añadimos limitaciones y futuras líneas de investigación.

Reviewer 2 Report

Impact and factorss associated with the caregivers of the people with physical disability might be very helpful to understand the impact of caregiving on their lives, develop the strategies for their role adaptation. But this manuscript didn’t tell a good story as the following concerns.

Major comments:

1、The title of the article "Impact and variables associated with the role of a caregiver of a person with a physical disability" did not match the results and methods of the research, and the results of the study only described psychological factors, which may cause misunderstanding to the readers.

2、In the systematic review, quality evaluation is an essential part of the study, and it is suggested to add the quality evaluation results. The search strategy of the study was not reasonable, and the literature search with the term “psychological characteristics*” was inappropriate and missed some psychological indicators.

3、The result of the study was not properly summarized, and the logical structure was not clear enough. The description of the psychological factors was too weak and did not present various factors in a good way. The result of the individual factor was too weak, only gender factors. The classification of sub-themes in the results of health effects was not reasonable. It is suggested that the result should be analyzed again.

4、The discussion section is too weak, and it is suggested that the discussion should closely follow the results of the study. The possible reasons for the results should be further analyzed. The results of this study should be better compared with other studies to make recommendations related to health and caregiving.

Minor comments:

1、The font in the flowchart was not uniform and the format was not standardized. It is suggested to refer to PRISMA flowchart and further optimize it.

2、It is suggested that the month of literature search should be added in the search strategy section.

Author Response

Response to Reviewer 2 Comments

Strong changes were made to substantially improve the manuscript, and we thank the reviewer for the keen observations. We feel that these changes were applied with the thoroughness expected by the reviewer, and again thank the reviewer for the review work.

Reviewer 2: Impact and factorss associated with the caregivers of the people with physical disability might be very helpful to understand the impact of caregiving on their lives, develop the strategies for their role adaptation. But this manuscript didn’t tell a good story as the following concerns. 

Authors: We would like to thank you for your job and your interest in this study.

Point 1: The title of the article "Impact and variables associated with the role of a caregiver of a person with a physical disability" did not match the results and methods of the research, and the results of the study only described psychological factors, which may cause misunderstanding to the readers. 

Response 1: Regarding the Title, your recommendations have been taken into account and we decided to restructure it:

Self-esteem, resilience, and impact on caregivers of a person with physical disability. Systematic Review

Point 2: In the systematic review, quality evaluation is an essential part of the study, and it is suggested to add the quality evaluation results. The search strategy of the study was not reasonable, and the literature search with the term “psychological characteristics*” was inappropriate and missed some psychological indicators. 

Response 2: Based on your suggestions, the methodology of the study was modified and the methodological limitations were added, explaining why we use the term "psychological characteristics*".

Point 3: The result of the study was not properly summarized, and the logical structure was not clear enough. The description of the psychological factors was too weak and did not present various factors in a good way. The result of the individual factor was too weak, only gender factors. The classification of sub-themes in the results of health effects was not reasonable. It is suggested that the result should be analyzed again. 

Response 3: The results were expanded and theoretically supported by other studies. The structure of the results was modified. Finally, the location of the studies, methodological aspects and biases in the sample selection, and reporting of the studies were included.

All changes have been marked in red.

Point 4: The discussion section is too weak, and it is suggested that the discussion should closely follow the results of the study. The possible reasons for the results should be further analyzed. The results of this study should be better compared with other studies to make recommendations related to health and caregiving.

Response 4: The discussion was supported by methodological aspects with respect to results found in other studies. More data related to health and care were included.

For example:

Based on the above, as shown by Gálvez et al. (2003), in recent years there has been an increase in the prevalence of mood disorders among caregivers of dependent people, especially increasing the frequency of anxiety and depression problems among them. It has been found that the higher the degree of dependency, the higher the prevalence of affective disorders among caregivers [81].

MINOR RESPONSE:

1、The font in the flowchart was not uniform and the format was not standardized. It is suggested to refer to PRISMA flowchart and further optimize it. 

2、It is suggested that the month of literature search should be added in the search strategy section. 

Thank you for your feedback. The PRISMA flowchart has been updated and the considerations provided have been modified.

Round 2

Reviewer 1 Report

The manuscript addresses a topic of interest to the international scientific community, such as the impact associated with caregiving and the variables present in caregivers of people with physical disabilities. In general, the evaluated study makes a significant contribution to the topic. 

The new version of the evaluated manuscript has made important improvements to the text (abstract, background, methodology, results, discussion and conclusions), following the recommendations appropriately. In this regard, the introduction provides sufficient background supported by a correct review of the international literature. In turn, the proposed design is appropriate for research. Furthermore, the methods are adequately described, and the results are clearly presented. Likewise, the discussion and conclusions are supported by the results. 

Finally, we consider that the new manuscript has been significantly improved and now guarantees publication.

Author Response

Response to Reviewer 1 Comments 

The manuscript addresses a topic of interest to the international scientific community, namely the impact associated with caregiving and the variables present in caregivers of people with physical disabilities. Overall, the peer-reviewed study makes a significant contribution to the topic.

The new version of the evaluated manuscript has made significant improvements in the text (abstract, background, methodology, results, discussion and conclusions), following the recommendations appropriately. In this sense, the introduction provides sufficient background supported by a proper review of the international literature. In turn, the proposed design is appropriate for the research. In addition, the methods are adequately described and the results are clearly presented. Furthermore, the discussion and conclusions are supported by the results.

Finally, we consider that the new manuscript has been significantly improved and now warrants publication.

Authors: Thank you very much for your job and for your positive assessment of the study.

Reviewer 2 Report

The impact and factors associated with the caregivers of the people with physical disability might be very helpful to understand the impact of caregiving on their lives, develop the strategies for their role adaptation. The following recommendations below need to be improved:

1、The title of the article "Self-esteem, resilience, and impact on caregivers of a person  with a physical disability. Systematic review" does not match the results. There is an error in punctuation in the title. It is suggested that the title should be further considered.

2、The result "3.3. Impact and psychological Characteristics Present in caregivers" still needs to be further improved, and it is suggested to present it in the form of sub-themes.

3、There is no further explanation of literature quality assessment in this manuscript, and it is suggested to use the quality evaluation tool.

Author Response

Response to Reviewer 2 Comments 

Comments and Suggestions for Authors

The impact and factors associated with the caregivers of the people with physical disability might be very helpful to understand the impact of caregiving on their lives, develop the strategies for their role adaptation. The following recommendations below need to be improved:

Authors: We would like to thank you for your job and your interest in this study. 

Point 1、The title of the article "Self-esteem, resilience, and impact on caregivers of a person  with a physical disability. Systematic review" does not match the results. There is an error in punctuation in the title. It is suggested that the title should be further considered.

Response 1: The new title for the paper is: Caregiving role, and psychosocial and individual factors: A systematic review", as it encompasses in higher order everything reviewed throughout this paper.

Point 2、The result "3.3. Impact and psychological Characteristics Present in caregivers" still needs to be further improved, and it is suggested to present it in the form of sub-themes.

Response 2:

The proposed sub-themes are:

3.2.1. Physical problems associated with caregiving.

3.2.2 Psychological and mental problems associated with caring.

3.2.2.1 Well-being and quality of life.

3.2.2.2 Emotional stress and psychological distress

3.2.2.3 Anxiety and depression.

3.2.2.4 Burden associated with caregiving.

3.2.3 Other problems associated with caregiving

3.2.4 Individual and protective factors

We have also thought it appropriate to improve the content of point 3.3.

Point 3、There is no further explanation of literature quality assessment in this manuscript, and it is suggested to use the quality evaluation tool.

Response 3:

Assessing the quality of the literature: Finally, the quality of each selected study was not assessed, because the systematic review conducted was essentially descriptive and not evaluative. This was for several reasons: first, it favoured the large coverage of studies retrieved and entered into the synthetic description of the studies. Second, the selected studies led to the observation and highlighting of possible biases in them, such as the induction of the reliability and validity of the measures used, among other things noted above. These specific limitations would have gone unnoticed if I had introduced a filter on these aspects. This does not indicate the absolute absence of quality assessment of the studies, because some assessment of this quality was partially guaranteed by the selection of the database (e.g., WoS, and Scopus), whose selective processes are high for choosing the journals receiving research articles.
